# Uncertainty Quantification and Deep Ensembles

Rahul Rahaman

Department of Statistics and Data Science,
National University of Singapore
`rahul.rahaman@u.nus.edu`

Alexandre H. Thiery

Department of Statistics and Data Science,
National University of Singapore
`a.h.thiery@nus.edu.sg`

## Abstract

Deep Learning methods are known to suffer from calibration issues: they typically produce over-confident estimates. These problems are exacerbated in the low data regime. Although the calibration of probabilistic models is well studied, calibrating extremely over-parametrized models in the low-data regime presents unique challenges. We show that deep-ensembles do not necessarily lead to improved calibration properties. In fact, we show that standard ensembling methods, when used in conjunction with modern techniques such as mixup regularization, can lead to less calibrated models. This text examines the interplay between three of the most simple and commonly used approaches to leverage deep learning when data is scarce: data-augmentation, ensembling, and post-processing calibration methods. Although standard ensembling techniques certainly help boost accuracy, we demonstrate that the calibration of deep ensembles relies on subtle trade-offs. We also find that calibration methods such as temperature scaling need to be slightly tweaked when used with deep-ensembles and, crucially, need to be executed *after* the averaging process. Our simulations indicate that this simple strategy can halve the Expected Calibration Error (ECE) on a range of benchmark classification problems compared to standard deep-ensembles in the low data regime.

## 1 Introduction

Overparametrized deep models can memorize datasets with labels entirely randomized [48]. It is consequently not entirely clear why such extremely flexible models are able to generalize well on unseen data and trained with algorithms as simple as stochastic gradient descent, although a lot of progress on these questions have recently been reported [8, 19, 2, 31, 39, 10].

The high capacity of neural network models, and their ability to easily overfit complex datasets, makes them especially vulnerable to calibration issues. In many situations, standard deep-learning approaches are known to produce probabilistic forecasts that are over-confident [16]. In this text, we consider the regime where the size of the training sets is very small, which typically amplifies these issues. This can lead to problematic behaviors when deep neural networks are deployed in scenarios where a proper quantification of the uncertainty is necessary. Indeed, a host of methods [22, 30, 40, 12, 37] have been proposed to mitigate these calibration issues, even though no gold standard has so far emerged. Many different forms of regularization techniques [35, 48, 50] have been shown to reduce overfitting in deep neural networks. Importantly, practical implementations and

35th Conference on Neural Information Processing Systems (NeurIPS 2021).

approximations of Bayesian methodologies [30, 44, 3, 14, 27, 38, 28] have demonstrated their worth in several settings. However, some of these techniques are not entirely straightforward to implement in practice. Ensembling approaches such as *drop-outs* [12] have been widely adopted, largely due to their ease of implementation. Recently, [1] provides a study on different ensembling techniques and describes pitfalls of certain metric for in-domain uncertainty quantification. Also subsequent to our work, several articles also studied the interaction between data-augmentation and calibration issues. Importantly, the CAMixup approach is proposed as a promising solution in [42]. Furthermore, [47] analyzes the under-confidence of ensembles due to augmentations from a theoretical perspective. In this text, we investigate the practical use of Deep-Ensembles [22, 4, 25, 41, 9, 16], a straightforward approach that leads to state-of-the-art performances in most regimes. Although deep-ensembles can be difficult to implement when training datasets are large (but calibration issues are less pronounced in this regime), the focus of this text is the data-scarce setting where the computational burden associated with deep-ensembles is not a significant problem.

**Contributions:** We study the interaction between three of the most simple and widely used methods for adopting deep-learning to the low-data regime: ensembling, temperature scaling, and mixup data augmentation.

- Despite the widely-held belief that model averaging improves calibration properties, we show that, in general, standard ensembling practices do not lead to better-calibrated models. Instead, we show that averaging the predictions of a set of neural networks generally leads to less confident predictions: that is generally only beneficial in the oft-encountered regime when each network is overconfident. Although our results are based on Deep Ensembles, our empirical analysis extends to any class of model averaging, including sampling-based Bayesian Deep Learning methods.

- We empirically demonstrate that networks trained with the *mixup* data-augmentation scheme, a widespread practice in computer vision, are typically under-confident. Consequently, subtle interactions between ensembling techniques and modern data-augmentation pipelines have to be considered for proper uncertainty quantification. The typical distributional shift induced by the mixup data-augmentation strategy influences the calibration properties of the resulting trained neural networks. In these settings, a standard ensembling approach typically worsens the calibration issues.

- Post-processing techniques such as *temperature scaling* are sometimes regarded as competing methods when comparing the performance of many modern model-averaging techniques. Instead, to mitigate the under-confidence of model averaging, temperature scaling should be used *in conjunction with* deep-ensembling methods. More importantly, the order in which the aggregation and the calibration procedures are carried out greatly influences the resulting uncertainty quantification. These findings lead us to formulate the straightforward *Pool-Then-Calibrate* strategy for post-processing deep-ensembles: **(1)** in a first stage, separately train deep models **(2)** in a second stage, fit a *single* temperature parameter by minimizing a proper scoring rule (eg. cross-entropy) on a validation set. In the low data regime, this simple procedure can halve the Expected Calibration Error (ECE) on a range of benchmark classification problems when compared to standard deep-ensembles. Although straightforward to implement, to the best of our knowledge this strategy has not been investigated in the literature prior to our work.

## 2 Background

Consider a classification task with $C \geq 2$ possible classes $\mathcal{Y} \equiv \{1, \ldots, C\}$. For a sample $x \in \mathcal{X}$, the quantity $\mathbf{p}(x) \in \Delta_C = \{\mathbf{p} \in \mathbb{R}_+^C : p_1 + \ldots + p_C = 1\}$ represents a probabilistic prediction, often obtained as $\mathbf{p}(x) = \sigma_{\mathrm{SM}}[\mathbf{f_w}(x)]$ for a neural network $\mathbf{f_w} : \mathcal{X} \to \mathbb{R}^C$ with weight $\mathbf{w} \in \mathbb{R}^D$ and softmax function $\sigma_{\mathrm{SM}} : \mathbb{R}^C \to \Delta_C$. We set $\widehat{y}(x) \equiv \arg \max_c p_c(x)$ and $\widehat{p}(x) = \max \mathbf{p}(x)$.

**Augmentation:** Consider a training dataset $\mathcal{D} \equiv \{x_i, y_i\}_{i=1}^N$ and denote by $\overline{y} \in \Delta_C$ the one-hot encoded version of the label $y \in \mathcal{Y}$. A stochastic augmentation process $\mathrm{Aug} : \mathcal{X} \times \Delta_C \to \mathcal{X} \times \Delta_C$ maps a pair $(x, \overline{y}) \in \mathcal{X} \times \Delta_C$ to another augmented pair $(x_\star, \overline{y}_\star)$. In computer vision, standard augmentation strategies include rotations, translations, brightness and contrast manipulations. In this text, in addition to these standard agumentations, we also make use of the more recently proposed *mixup* augmentation strategy [49] that has proven beneficial in several settings. For a pair

$(x, \overline{y}) \in \mathcal{X} \times \Delta_C$, its mixup-augmented version $(x_\star, \overline{y}_\star)$ is defined as

$$x_\star = \gamma\,x + (1-\gamma)\,x_J \quad \text{and} \quad \overline{y}_\star = \gamma\,\overline{y} + (1-\gamma)\,\overline{y}_J$$

for a random coefficient $\gamma \in (0,1)$ drawn from a fixed mixing distribution often chosen as $\mathrm{Beta}(\alpha, \alpha)$, and a random index $J$ drawn uniformly within $\{1, \dots, N\}$.

**Model averaging:** Ensembling methods leverage a set of models by combining them into an aggregated model. In the context of deep learning, Bayesian averaging consists of weighting the predictions according to the Bayesian posterior $\pi(d\mathbf{w} \mid \mathcal{D}_{\text{train}})$ on the neural weights. Instead of finding an optimal set of weights by minimizing a loss function, predictions are averaged. Denoting by $\mathbf{p_w}(x) \in \Delta_C$ the probabilistic prediction associated to sample $x \in \mathcal{X}$ and neural weight $\mathbf{w}$, the Bayesian approach advocates to consider

$$(\text{prediction}) \equiv \int \mathbf{p_w}(x)\,\pi(d\mathbf{w} \mid \mathcal{D}_{\text{train}}) \in \Delta_C. \tag{1}$$

Designing sensible prior distributions is still an active area of research, and data-augmentation schemes, crucial in practice, are not entirely straightforward to fit into this framework. Furthermore, the high-dimensional integral (1) is (extremely) intractable: the posterior distribution $\pi(d\mathbf{w}|\mathcal{D}_{\text{train}})$ is multi-modal, high-dimensional, concentrated along low-dimensional structures, and any local exploration algorithm (eg. MCMC, Langevin dynamics and their variations) is bound to only explore a tiny fraction of the state space. Because of the typically large number of degrees of symmetries, many of these local modes correspond to essentially similar predictions, indicating that it is likely not necessary to explore all the modes in order to approximate (1) well. A detailed understanding of the geometric properties of the posterior distribution in Bayesian neural networks is still lacking, although a lot of recent progress has been made. Indeed, variational approximations have been reported to improve, in some settings, over standard empirical risk minimization procedures. Deep-ensembles can be understood as crude, but practical, approximations of the integral in Equation (1). The high-dimensional integral can be approximated by a simple non-weighted average over several modes $\mathbf{w}_1, \dots, \mathbf{w}_K$ of the posterior distribution found by minimizing the negative log-posterior, or some approximations of it, with standard optimization techniques:

$$(\text{prediction}) \equiv \frac{1}{K}\Big\{ \mathbf{p}_{\mathbf{w}_1}(x) + \dots + \mathbf{p}_{\mathbf{w}_K}(x) \Big\} \in \Delta_C.$$

We refer the interested reader to [34, 29, 45, 3] for different perspectives on Bayesian neural networks. Although simple and not well understood, deep-ensembles have been shown to provide highly robust uncertainty quantification when compared to more sophisticated approaches [22, 4, 25, 41].

**Post-processing Calibration Methods:** The article [16] proposes a class of post-processing calibration methods that extend the more standard *Platt Scaling* approach [36]. *Temperature Scaling*, the simplest of these methods, transforms the probabilistic outputs $\mathbf{p}(x) \in \Delta_C$ into a tempered version $\mathrm{Scale}[\mathbf{p}(x), \tau] \in \Delta_C$ defined through the scaling function

$$\mathrm{Scale}(\mathbf{p}, \tau) \equiv \sigma_{\text{SM}}\big(\log \mathbf{p}/\tau\big) = \frac{1}{Z}\left( p_1^{1/\tau}, \dots, p_C^{1/\tau} \right) \in \Delta_C, \tag{2}$$

for a temperature parameter $\tau > 0$ and normalization $Z > 0$. The optimal parameter $\tau_\star > 0$ is usually found by minimizing proper-scoring rules [13], often chosen as the negative log-likelihood, on a validation dataset. Crucially, during this post-processing step, the parameters of the probabilistic model are kept fixed: the only parameter being optimized is the temperature $\tau > 0$. In the low-data regime, the validation set being also extremely small, we have empirically observed that the more sophisticated *Vector* and *Matrix* scaling post-processing calibration methods [16] do not offer any significant advantage over temperature scaling approach and in fact overfit the extremely small validation dataset as chosen by our setup.

**Calibration Metrics:** The *Expected Calibration Error* (ECE) measures the discrepancy between prediction confidence and empirical accuracy. For a partition $0 = c_0 < \dots < c_M = 1$ of the unit interval and a labelled set $\{x_i, y_i\}_{i=1}^N$, set $B_m = \{i : c_{m-1} < \widehat{p}(x_i) \le c_m\}$. The quantity ECE is then defined as

$$\mathrm{ECE} = \sum_{m=1}^M \frac{|B_m|}{N}\big|\,\mathrm{conf}_m - \mathrm{acc}_m\,\big| \tag{3}$$

$$\text{where} \quad \mathrm{acc}_m = \frac{1}{|B_m|}\sum_{i \in B_m} \mathbf{1}(\widehat{y}(x_i) = y_i) \quad \text{and} \quad \mathrm{conf}_m = \frac{1}{|B_m|}\sum_{i \in B_m} \widehat{p}(x_i). \tag{4}$$

A model is calibrated if $\mathrm{acc}_m \approx \mathrm{conf}_m$ for all $1 \leq m \leq M$. It is often instructive to display the associated *reliability curve*, i.e. the curve with $\mathrm{conf}_m$ on the x-axis and the difference $(\mathrm{acc}_m - \mathrm{conf}_m)$ on the y-axis. Figure 1 displays examples of such reliability curves. A perfectly calibrated model is flat (i.e. $\mathrm{acc}_m - \mathrm{conf}_m = 0$), while the reliability curve associated to an under-confident (resp. over-confident) model prominently lies above (resp. below) the flat line $\mathrm{acc}_m - \mathrm{conf}_m = 0$. We sometimes also report the value of the Brier score [5] defined as $\frac{1}{N}\sum_{i=1}^{N} \|\mathbf{p}(x_i) - \overline{y}_i\|_2^2$.

**Setup and implementation details:** For our experiments, we use standard neural architectures. For CIFAR10/100 [21] we use ResNet18, ResNet34 [17] for Imagenette/Imagewoof [18], and for the Diabetic Retinopathy [7], similar to [26] we use the architecture (not containing any residual connection) from the $5^{th}$ place solution of the associated *Kaggle* challenge. We also include the results for LeNet [23] trained on the MNIST [24] dataset in the supplementary. A very low number of training examples (CIFAR10: 1000, CIFAR100: 5000, Image{nette, woof}: 5000, MNIST: 500) was used for all the datasets. However, we also show that our observations extend to full-data setups in 4. The validation dataset is chosen from the leftover training dataset. The test dataset is kept as the original and is hidden during both training and validation step.

## 3   Empirical Observations

**Linear pooling:** It has been observed in several studies that averaging the probabilistic predictions of a set of independently trained neural networks, i.e., deep-ensembles, often leads to more accurate and better-calibrated forecasts [22, 4, 25, 41, 9]. Figure 1 displays the reliability curves across three different datasets of a set of $K = 30$ independently trained neural networks, as well as the reliability curves of the aggregated forecasts obtained by simply linear averaging the $K = 30$ individual probabilistic predictions. These results suggest that deep-ensembles consistently lead to predictions that are *less confident* than the ones of its individual constituents. This can indeed be beneficial in the often encountered situation when each individual neural network is overconfident. Nevertheless, this phenomenon should not be mistaken with an intrinsic property of deep ensembles to lead to better-calibrated forecasts. For example, and as discussed further in Section 4, networks trained with the popular *mixup* data-augmentation are typically under-confident. Ensembling such a set of individual networks typically leads to predictions that are even more under-confident.

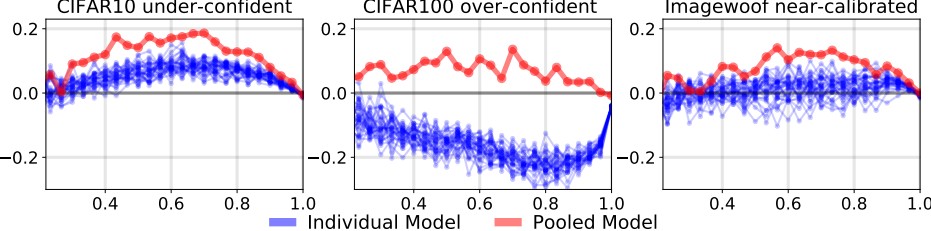

Figure 1: **Confidence** $\mathrm{conf}_m$ **(x-axis) vs. Difference** $(\mathrm{acc}_m - \mathrm{conf}_m)$ **(y-axis):** We plot *Reliability Curves* in this figure, see Section 2 'Calibration Metrics' for definitions. The plots display the curves of $K = 30$ individual networks (**blue**) trained on three datasets (i.e. CIFAR10, CIFAR100 and Imagewoof), as well as the pooled estimates (**red**) obtained by averaging the $K$ individual predictions. This linear averaging leads to consistently less confident predictions (i.e. higer values of $(\mathrm{acc}_m - \mathrm{conf}_m)$). It is only beneficial to calibration when each network is over-confident. It is typically detrimental to calibration when the individual networks are already calibrated, or under-confident.

**Other BNN methods:** It is important to point out that under-confidence of pooled predictions are not limited to Deep Ensembles. Other modern Bayesian Neural Network methods show similar properties. In table 1 we can see that ensembles obtained by SWAG [30] and MC-Dropout [11], two other popular model averaging techniques, are more under-confident than the individual models.

In order to gain some insights into this phenomenon, recall the definition of the entropy functional $\mathcal{H}: \Delta_C \to \mathbb{R}$, defined as $\mathcal{H}(\mathbf{p}) = -\sum_{k=1}^{C} p_k \log p_k$. The entropy functional is concave on the probability simplex $\Delta_C$, i.e. $\mathcal{H}(\lambda \mathbf{p} + (1-\lambda)\mathbf{q}) \geq \lambda\,\mathcal{H}(\mathbf{p}) + (1-\lambda)\,\mathcal{H}(\mathbf{q})$ for any $\mathbf{p}, \mathbf{q} \in \Delta_C$. Furthermore, tempering a probability distribution $\mathbf{p}$ leads to an increased entropy if $\tau > 1$, as can

| Dataset | Method | Single models | Ensemble |
|---------|--------|---------------|----------|
| CIFAR 10 | SWAG | $3.17 \pm .27$ | 4.36 |
| | MC-Dropout | $6.55 \pm .10$ | 7.59 |
| CIFAR 100 | SWAG | $3.34 \pm .14$ | 5.49 |
| | MC-Dropout | $4.92 \pm .19$ | 9.05 |

Table 1: ECE, as defined in Equation (3), of twenty individual models and the ensemble of SWAG [30] and MC-Dropout [11] trained with mixup augmentation on full CIFAR{10,100} dataset. The ensemble is less calibrated than the individual models.

be proved by examining the derivative of the function $\tau \mapsto \mathcal{H}[\mathbf{p}^{1/\tau}]$. The entropy functional is consequently a natural surrogate measure of (lack of) confidence. The concavity property of the entropy functional shows that ensembling a set of $K$ individual networks leads to predictions whose entropies are higher than the average of the entropies of the individual predictions. We provide additional quantitative heuristics for this phenomenon in the *Supplementary Material*.

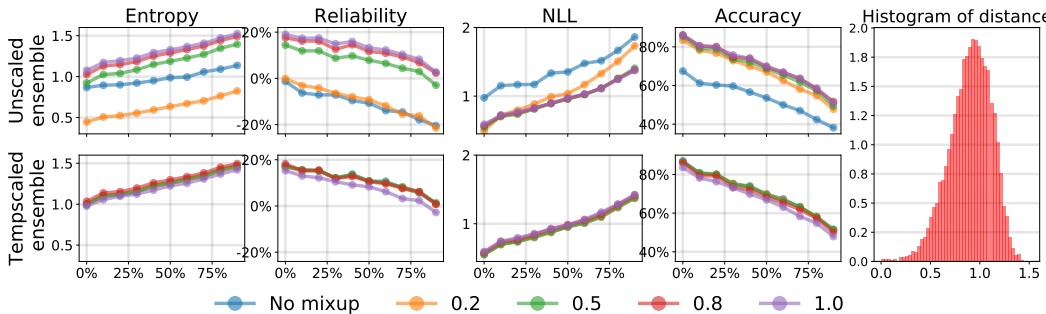

Figure 2: **Metric (y-axis) vs. Distance from the training set (percentile) (x-axis)**: Deep Ensembles trained on $N = 1000$ CIFAR10 samples with different amount of mixup regularization $\alpha \in \{.2, .5, .8, 1\}$. The x-axis represents quantiles of the distance to the CIFAR10 training set (see Section 3 for details). The overall distribution of the distances is displayed in the last column. The first row describes the performances of standard Deep Ensembles trained with data-augmentation and several amounts of mixup regularization $\alpha$. In the second row, before averaging the predictions of the members of the ensemble, each individual network is first temperature scaled on a validation set of size $N_{\text{val}} = 50$: this corresponds to method (**B**) of Section 4.

**Distance to the training set:** In order to gain some additional insights into the calibration properties of neural networks trained on small datasets, as well as the influence of the popular mixup augmentation strategy, we examine several metrics (i.e., Accuracy, Reliability, Negative Log-likelihood (NLL), Entropy) as a function of the distance to the (small) training set $\mathcal{D}_{\text{train}}$. The 2nd column of Figure 2 displays the mean *Reliability* (i.e., $\text{acc} - \text{conf}$) as a function of the distance percentiles. We focus on the CIFAR10 dataset and train our networks on a balanced subset of $N = 1000$ training examples. Since there is no straightforward and semantically meaningful distance between images, we first use an unsupervised method (i.e., labels were not used) for learning a low-dimensional and semantically meaningful representation of dimension $d = 128$. For these experiments, we obtained a mapping $\Phi : \mathbb{R}^{32,32} \rightarrow \mathbf{S}^{128}$, where $\mathbf{S}^{128} \subset \mathbb{R}^{128}$ denotes the unit sphere in $\mathbb{R}^{128}$, with the *SimCLR* method [6]. We used the distance $d(x, y) = \|\Phi(x) - \Phi(y)\|_2$, which in this case is equivalent to the cosine distance between the 128-dimensional representations of the CIFAR10 images $x$ and $y$. The distance of a test image $x$ to the training dataset is defined as $\min\{d(x, y_i) : y_i \in \mathcal{D}_{\text{train}}\}$. We computed the distances to the training set for each image contained in the standard CIFAR10 test set (last column of Figure 2). Not surprisingly, we note that the average Entropy, Negative Log-likelihood, and Error Rate all increase for test samples further away from the training set.

- **Over-confidence:** The second column represents the *Reliability curve*, but with bins (x-axis) as distance percentile, rather than confidence. The predictions associated with samples chosen further away from the training set have a lower value of $\text{acc} - \text{conf}$. This indicates that the *over*-confidence of the predictions increases (esp. lower mixup $\alpha$) with the distance

to the training set. In other words, even if the entropy increases as the distance increases (as it should), calibration issues do not vanish as the distance to the training set increases. This phenomenon is irrespective of the amount of mixup used for training the network.

- **Effect of mixup-augmentation:** The first row of Figure 2 shows that increasing the amount of mixup augmentation generally leads to an increase in entropy, decrease in over-confidence, as well as more accurate predictions (lower NLL and higher accuracy). Additionally, the effect is less pronounced for $\alpha \geq 0.2$. This is confirmed in Figure 3 that displays more generally the effect of the mixup-augmentation on the reliability curves over four different datasets. In the supplementary we provide more analysis on this.

- **Temperature Scaling:** Importantly, the second row of Figure 2 indicates that a post-processing temperature scaling for the individual models almost washes-out all the differences due to the mixup-augmentation scheme. For this experiment, an ensemble of $K = 30$ networks is considered: before averaging the predictions, each network has been individually temperature scaled by fitting a temperature parameter (through negative likelihood minimization) on a validation set of size $N_{\text{valid}} = 50$.

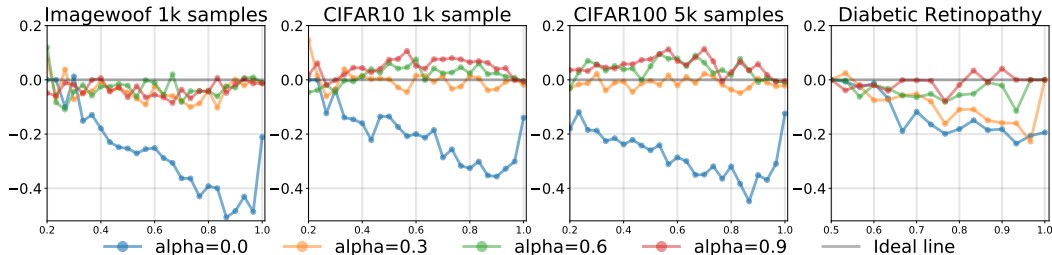

Figure 3: **Confidence** $\text{conf}_m$ **(x-axis) vs. Difference** $(\text{acc}_m - \text{conf}_m)$ **(y-axis):** *Reliability curve* of a single neural network trained with different amount of mixup-augmentation on the Imagewoof, CIFAR10, CIFAR100 and Diabetic Retinopathy datasets. Increasing the amount of mixup augmentation, in general, makes the predictions *less*-confident. The case $\alpha = 0$ corresponds to training without an mixup-augmentation, i.e. only using standard augmentation strategies.

## 4 Calibrating Deep Ensembles

In order to calibrate deep ensembles, several methodologies can be considered:

**(A)** Do nothing and hope that the averaging process intrinsically leads to better calibration

**(B)** Calibrate each individual network before aggregating all the results

**(C)** Simultaneously aggregate and calibrate the probabilistic forecasts of each individual model.

**(D)** Aggregate first the estimates of each individual model before calibrating the pooled estimate.

Simple pooling/aggregation rules that do not require a large number of tuning parameters are usually preferred, especially when training data is scarce [20, 46]. Such rules are usually robust, conceptually easy to understand, and straightforward to implement and optimize. The standard and most commonly used average pooling of a set $\mathbf{p}^{1:K}$ of $K \geq 2$ probabilistic predictions $\mathbf{p}^{(1)}, \ldots, \mathbf{p}^{(K)} \in \Delta_C \subset \mathbb{R}^C$ is defined as

$$\mathbf{Agg}(\mathbf{p}^{1:K}) = \frac{\mathbf{p}^1 + \ldots + \mathbf{p}^K}{K}. \tag{5}$$

Replacing the averaging with the median operation leads to *median pooling* strategy, where the median is taken component-wise and then normalized afterward to obtain the final probability prediction. Alternatively, *trimmed linear pooling* strategy removes a pre-defined percentage of outlier predictions before performing the average in 5.

**Pool-Then-Calibrate (D):** any of the aforementioned aggregation procedure can be used as a pooling strategy before fitting a temperature $\tau_\star$ by minimizing proper scoring rules on a validation set. In all

our experiments, we minimized the negative log-likelihood (i.e., cross-entropy). For a given set $\mathbf{p}^{1:K}$ of $K \geq 2$ probabilistic forecasts, the final prediction is defined as

$$\mathbf{p}_\star \equiv \text{Scale}\left[\mathbf{Agg}(\mathbf{p}^{1:K}), \tau_\star\right] \tag{6}$$

where $\text{Scale}(\mathbf{p}, \tau) \equiv \sigma_{\text{SM}}\left(\log \mathbf{p}/\tau\right)$. Note that the aggregation procedure can be carried out entirely independently from the fitting of the optimal temperature $\tau_\star$.

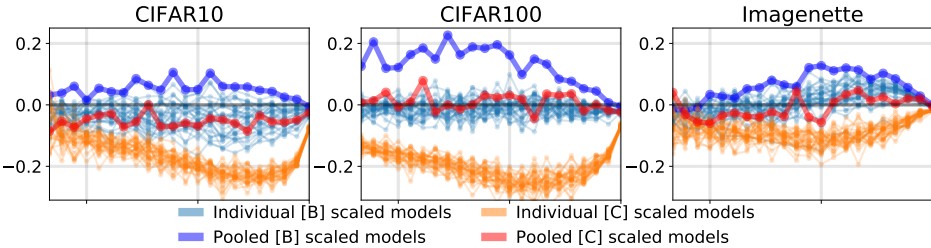

Figure 4: **Confidence** $\text{conf}_m$ **(x-axis) vs. Difference** $(\text{acc}_m - \text{conf}_m)$ **(y-axis):** Reliability curve of: **(light blue)** each model calibrated with one temperature per model (i.e. individually temperature scaled), **(dark blue)** average of individually temperature scaled models (i.e. method **[B]**), **(orange)** each model scaled with a global temperature obtained with method **[C]**, **(red)** result of method **[C]** that consists in simultaneously aggregating and calibrating the probabilistic forecasts of each individual model. **Datasets:** a train:validation split of size $950 : 50$ was used for the CIFAR10 and IMAGENETTE datasets, and of size $4700 : 300$ for the CIFAR100 dataset. To avoid clutter, we omit method **[D]** for its similarity with method **[C]** in terms of performance.

**Joint Pool-and-Calibrate (C):** there are several situations when the so-called *end-to-end* training strategy consisting in jointly optimizing several component of a composite system leads to increased performances [33, 32, 15]. In our setting, this means learning the optimal temperature $\tau_\star$ concurrently with the aggregation procedure. The optimal temperature $\tau_\star$ is found by minimizing a proper scoring rule $\text{Score}(\cdot)$ on a validation set $\mathcal{D}_{\text{valid}} \equiv \{x_i, y_i\}_{i=1}^{N_{\text{val}}}$,

$$\tau_\star = \arg\min\left\{\tau \mapsto \frac{1}{\mathcal{D}_{\text{valid}}} \sum_{i \in \mathcal{D}_{\text{valid}}} \text{Score}(\mathbf{p}_i^\tau, y_i)\right\}, \tag{7}$$

where $\mathbf{p}_i^\tau = \mathbf{Agg}\left[\text{Scale}(\mathbf{p}^{1:K}(x_i), \tau)\right] \in \Delta_C$ denotes the aggregated probabilistic prediction for sample $x_i$. In all our experiments, we have found it computationally more efficient and robust to use a simple grid search for finding the optimal temperature; we used $n = 100$ temperatures equally spaced on a logarithmic scale in between $\tau_{\min} = 10^{-2}$ and $\tau_{\max} = 10$.

**Importance of the Pooling and Calibration order:** Figure 4 shows calibration curves when individual models are temperature scaled separately (i.e. group **[B]** of methods), as well as when the models are scaled with a common temperature parameter (i.e. group **[C]** of methods). Furthermore, the calibration curves of the pooled model (group **[B]** and **[C]** of methods) are also displayed. More formally, the group **[B]** of methods obtains for each individual model $1 \leq k \leq K$ an optimal temperature $\tau_\star^{(k)} > 0$ as solution of the optimization procedure

$$\tau_\star^{(k)} = \arg\min_\tau \frac{1}{\mathcal{D}_{\text{valid}}} \sum_{i \in \mathcal{D}_{\text{valid}}} \text{Score}\left(\text{Scale}\left[\mathbf{p}_i^k, \tau\right], y_i\right)$$

where $\mathbf{p}_i^k \in \Delta_C$ denotes the probabilistic output of the $k^{th}$ model for the $i^{th}$ example in validation dataset. The *light blue* calibration curves corresponds to the outputs $\text{Scale}\left[\mathbf{p}^k, \tau_\star^{(k)}\right]$ for $K$ different models. The *deep blue* calibration curve corresponds the linear pooling of the individually scaled predictions. For the group **[C]** of methods, a single common temperature $\tau_\star > 0$ is obtained as solution of the optimization procedure described in equation (7). The *orange* calibration curves are generated using the predictions $\text{Scale}\left[\mathbf{p}^k, \tau_\star\right]$, and the *red* curve corresponds to the prediction $\mathbf{Agg}\left[\text{Scale}(\mathbf{p}^{1:K}, \tau_\star)\right]$. Notice that when scaled separately (by $\tau_\star^{(k)}$), each of the individual models (light blue) is close to being calibrated, but the resulting pooled model (deep blue) is under-confident. However, when scaled by a common temperature, the optimization chooses a temperature $\tau_\star$ that

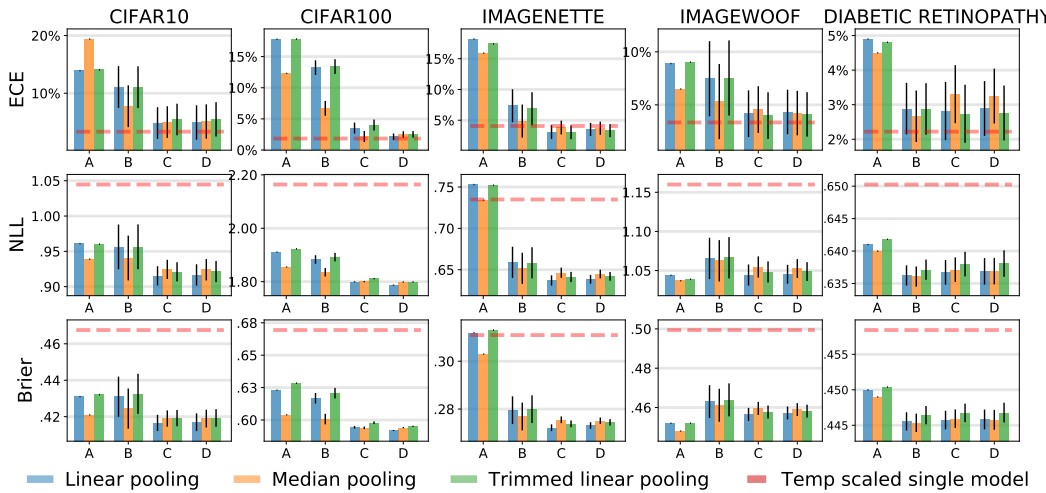

Figure 5: **Pooling method (x-axis) vs. Metric (y-axis):** Performance of different pooling strategies (**A-D**) with $K = 30$ models trained with mixup-augmentation ($\alpha = 1$) across multiple datasets. The total datasets (training + validation) were of size $N = 1000$ for CIFAR10 and Imagenette and Imagewoof, and $N = 5000$ for CIFAR100 and Diabetic Retinopathy. Experiments were executed 50 times on the same training data but different validation sets. The dashed red line represents a baseline performance when a single model was training with mixup augmentation ($\alpha = 1$) and post-processed with temperature scaling.

makes the individual models (orange) slightly over-confident so that the resulting pooled model (red) is nearly calibrated. This reinforces the justifications in section 3, and it also shows the importance of the order of pooling and scaling.

Figure 5 compares the four methodologies **A-B-C-D** identified at the start of this section, with the three different pooling approaches **Agg**$_{\text{avg}}$ and **Agg**$_{\text{med}}$ and **Agg**$_{\text{trim}}$. These methods are compared to the baseline approach (in dashed red line) consisting of fitting a single network trained with the same amount $\alpha = 1$ of mixup augmentation before being temperature scaled. All the experiments are executed 50 times, on the *same* training set, but with 50 different validation sets of size $N_{\text{val}} = 50$ for CIFAR10, Imagenette, Imagewoof and $N_{\text{val}} = 300$ for CIFAR100, and $N_{\text{val}} = 500$ for the Diabetic Retinopathy dataset. The results indicate that on most metrics and datasets, the (naive) method (**A**) consisting of simply averaging predictions is not competitive. Secondly, and as explained in the previous section, the method (**B**) consisting in first calibrating the individual networks before pooling the predictions is less efficient across metrics than the last two methods (**C** − **D**). Finally, the two methods (**C** − **D**) perform comparably, the method (**D**) (i.e. *pool-then-calibrate*) being slightly more straightforward to implement. With regards to the pooling methods, the intuitive robustness of the *median* and *trimmed-averaging* approaches do not seem to lead to any consistent gain across metrics and datasets. Note that ensembling a set of $K = 30$ networks (without any form of post-processing) does lead to a very significant improvement in NLL and Brier score but leads to a serious deterioration of the ECE. The *Pool-Then-Calibrate* keeps the gains in NLL/Brier score unaffected, without compromising calibration.

**Importance of the validation set:** it would be practically useful to be able to fit the temperature without relying on a validation set. We report that using the training set instead (obviously) does not lead to better-calibrated models. We have tried to use a different amount of mixup-augmentation (and other types of augmentation) on the training set for fitting the temperature parameter but have not been able to obtain satisfying results.

**Role and effect of mixup-augmentation:** the mixup augmentation strategy is popular and straightforward to implement. As already empirically described in Section 3, increasing the amount of mixup-augmentation typically leads to a decrease in the confidence and increase in entropy of the predictions. This can be beneficial in some situations but also indicates that this approach should

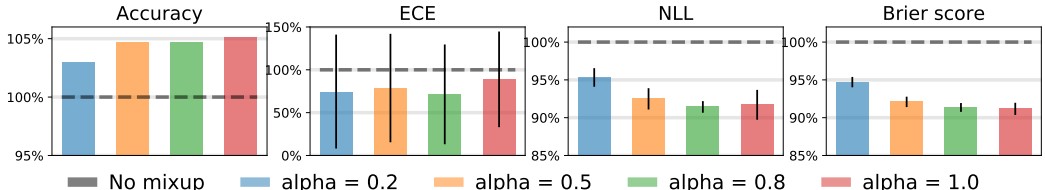

Figure 6: **Mixup strength (x-axis) vs. Metric (y-axis):** *Pool-Then-Calibrate* approach when applied to a deep-ensemble of $K = 30$ networks trained with different amount of mixup-augmentation on $N = 1000$ CIFAR10 training samples ($N_{\text{val}} = 50$ were used for validation). For each metric, we report the **ratio** of performance when compared to the *Pool-then-Calibrate* method used without any form of mixup-augmentation (only standard data-augmentation). The results indicate a clear benefit in using the mixup-augmentation in conjunction with temperature scaling. The error bars represent variability due to the choice of different validation sets.

certainly be employed with care for producing calibrated probabilistic predictions. Contrarily to other geometric data-augmentation transformations such as image flipping, rotations, and dilatations, the mixup strategy produces non-realistic images that consequently lie outside the data-manifold of natural images: leading to a large distributional shift. Mixup relies on a subtle trade-off between the increase in training data diversity, which can help mitigate over-fitting problems, and the distributional shift that can be detrimental to the calibration properties of the resulting method. Figure 6 compares the performance of the *Pool-Then-Calibrate* approach when applied to a deep ensemble of $K = 30$ networks trained with different amounts of mixup-$\alpha$. The results are compared to the same approach (i.e. *Pool-then-Calibrate* with $K = 30$ networks) with no mixup-augmentation. The results indicate a clear benefit in using the mixup-augmentation in conjunction with temperature scaling.

**Extension to full-data setting:** Although classification accuracy is usually not an issue when data is plentiful, the lack of calibration can indeed be still present when models are trained with aggressive data-augmentation strategies (as is common nowadays): the distributional shift between (data-augmented) training samples and (non-augmented) test samples when models are used in production can lead to significant calibration issues. Although we mainly focus on low-data setting, below in table 2 we show that our conclusion extends to full-data setting as well. We have investigated below the CIFAR100 full dataset (ResNet architecture / no-mixup) setting under varying conditions.

| Method | Accuracy | ECE | NLL | Brier |
|---|---|---|---|---|
| (1) Individual models (unscaled) | $70.8 \pm .36$ | $9.8 \pm .31$ | $1.17 \pm .01$ | 0.411 |
| Ensemble of models in (1) | 78.4 | 5.9 | 0.782 | 0.308 |
| (2) Individual models (temp scaled) | $70.8 \pm .36$ | $2.1 \pm .4$ | $1.07 \pm .01$ | 0.396 |
| Ensemble of models in (2) | 78.4 | 13.2 | 0.859 | 0.331 |
| Pool-then-calibrate | 78.4 | 3.4 | 0.770 | 0.303 |

Table 2: In line with our discussion in Sec 3, we show that linear pooling **(A)** appears to be helping with calibration ($2^{nd}$ row) when individual models are mildly over-confident ($1^{st}$ row), but performs worse ($4^{th}$ row) than individual models even in full-data setting (CIFAR100 50K training) when the individual models are near-calibrated ($3^{rd}$ row). Our proposed *pool-then-calibrate* **(D)** has the best performance ($5^{th}$ row).

The first row reports the performance of individual models trained without mixup: the individual models are over-confident, but not extremely over-confident (presumably because of the large number of samples). When these models are pooled to make an ensemble in the second row, the pooled model is better calibrated. This is the setup that is usually studied in almost every early articles investigating the properties of deep-ensembles, hence leading to the conclusion that deep-ensembling inherently brings calibration. When we make the individual models calibrated in the $3^{rd}$ row, where we used temp-scaling but it can also be due to the effect of more aggressive data-augmentation schemes, the individual calibration naturally improves significantly. Nevertheless, when we pool these calibrated models to make an ensemble, the pooled model suffers from extreme under-confidence ($4^{th}$ row). Our proposed method *pool-then-calibrate* ($5^{th}$ row) performs well even in full-data setting.

**Out-of-distribution performance:** We show the out-of-distribution detection performance of our method compared to vanilla ensembling when the ensembles are trained on CIFAR10 and tested on a subset of CIFAR100 classes which are visually different from CIFAR10. In table 3, we show the metric: difference between the medians of the in-class and out-of-class prediction entropy (higher is better).

| Single model 30 variations | Deep Ensemble [A] | Pool-then-calibrate [D] |
|:---:|:---:|:---:|
| $0.342 \pm 0.015$ | 0.359 | 0.521 |

Table 3: Difference in median prediction entropy between in-class (CIFAR10) vs out-of-class (CIFAR100 subset) dataset. Pool-then-calibrate brings significant improvement in terms of out-of-distribution detection.

Pool-then-Calibrate performs significantly better than vanilla ensemble in separating the predictions for in-class and out-of-class observations (45% more separation in terms of distance between medians). In table 4, we also show the performance when we run inference on the CIFAR10-C dataset (Gaussian noise) after training our ensemble model on the setting: 1000 samples of CIFAR10 dataset with mixup 1.0. As expected, vanilla ensembling with linear pooling (**A**) has worse calibration than single models, while pool-then-calibrate (**D**) improves score across the board.

| Method | Accuracy | ECE | NLL | Brier |
|:---:|:---:|:---:|:---:|:---:|
| Individual models | $59.38 \pm 0.05$ | $6.57 \pm 0.006$ | $1.237 \pm 0.012$ | $0.549 \pm 0.005$ |
| Deep Ensemble **[A]** | 64.63 | 15.13 | 1.145 | 0.511 |
| Pool-then-Calibrate **[D]** | 64.63 | 1.65 | 1.059 | 0.480 |

Table 4: Inference on CIFAR-10C (Gaussian noise), trained on CIFAR10 (1K sample). Pool-then-calibrate **[D]** performs better while vanilla ensemble **[A]** has worse calibration than single models.

**Additional experiments:** In the supplementary, we add more experiments on the effect of number of models in the ensemble, detailed numerical results for all datasets as well as MNIST, ablation study, and effect of different mixup levels on all the metrics.

**Cold posteriors:** the article [43] reports gains in several metrics when fitting Bayesian neural networks to a *tempered posterior* of type $\pi_\tau(\theta) \propto \pi(\theta)^{1/\tau}$, where $\pi(\theta)$ is the standard Bayesian posterior, for temperatures $\tau$ *smaller than one*. Although not identical to our setting, it should be noted that in all our experiments, the optimal temperature $\tau_\star$ was consistently smaller than one. In our setting, this is because simply averaging predictions lead to under-confident results. We postulate that related mechanisms are responsible for the observations reported in [43].

## 5   Discussion

The problem of calibrating deep-ensembles has received surprisingly little attention in the literature. In this text, we examined the interaction between three of the most simple and widely used methods for adopting deep-learning to the low-data regime: ensembling, temperature scaling, and mixup data augmentation. We highlight that ensembling in itself does not lead to better-calibrated predictions, that the mixup augmentation strategy is practically important and relies on non-trivial trade-offs, and that these methods subtly interact with each other. Crucially, we demonstrate that the order in which the pooling and temperature scaling procedures are executed is important to obtaining calibrated deep-ensembles. We advocate the *Pool-Then-Calibrate* approach consisting of first pooling the individual neural network predictions together before eventually post-processing the result with a simple and robust temperature scaling step.

## 6   Broader Impact

Producing well-calibrated probabilistic predictions is crucial to risk management, and when decisions that rely on the outputs of probabilistic models have to be trusted. Furthermore, designing well-calibrated models is crucial to the adoption of machine-learning methods by the general public, especially in the field of AI-driven medical diagnosis, since it is intimately related to the issue of trust in new technologies.

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
