# 1 Additional experiments

**Out of sample detection**  We trained an ensemble of 30 ResNet18 models on 1000 CIFAR10 samples and evaluated our model on **(I)** CIFAR10 test examples (*In-class*) **(II)** a subset of Cifar100 test examples (*Out-of-class*) chosen to be visually different from the CIFAR10 training samples (e.g. apple, dinosaur, mountain, shark, skyscraper). Table 1 reports the difference between the median entropy of the predictions for the *Out-of-class* and *In-class* samples. Our proposed methodology leads to an increased discrepancy (ie. better *Out-of-class* detection) between the *Out-of-class* and *In-class* samples when compared to standard deep-ensembles.

| Single model 30 variations | Deep Ensemble method [A] | *Pool-then-calibrate* method [D] |
|---|---|---|
| $0.342 \pm 0.015$ | 0.359 | **0.521** |

Table 1: Difference between the median entropy of *in-class* and *out-of-class*. The *in-class* is chosen to be CIFAR10 test dataset and the *out-of-class* is a subset of CIFAR100 test dataset that has no visible similarity with CIFAR10 classes. We train an ensemble of 30 models on (CIFAR10 / 1000 samples / mixup $\alpha$ 1.0 / ResNet18) and then evaluate on the describe *in-class* vs. *out-of-class* datasets.

**Effect of varying the size of the ensemble**  In order to see whether our observations hold for a varying number of models in the ensemble pool, we plot different metrics for different numbers of models in the pool. Figure 1 reports the performance of the method *Calibrate-then-pool* [B] and the method *Pool-then-calibrate* [D] as a function of the ensemble size (average of 10 experiments are reported) We do not show the performance of vanilla pooling [A] in this figure. For that purpose we use table 2 to report the calibration performances for different ensembles size (CIFAR10 / 1000 samples / mixup $\alpha$ 1.0 / ResNet18).

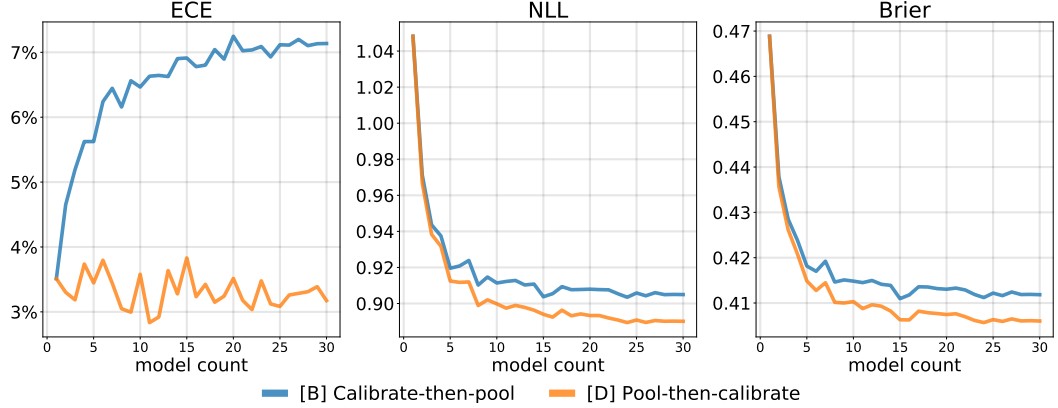

Figure 1: **Number of model (x-axis) vs. Metric (y-axis):** Comparison of the methods *Calibrate-then-pool* [B] and *Pool-then-calibrate* [D] described in Section 4 of the main text, on the CIFAR10 dataset with $N = 1000$ samples (950:50 split). The $x$-axis denotes the size of the ensemble. To avoid clutter and due to significantly worse performance, method [A] (i.e. vanilla averaging without scaling) is omitted. Averages over 10 expriments are reported.

| | Num model | 1 | 4 | 8 | 15 |
|---|---|---|---|---|---|
| Deep Ensemble method [A] | ECE | 7.31 | 12.37 | 13.44 | 13.87 |
| | Brier | 0.464 | 0.440 | 0.435 | 0.432 |
| *Pool-then-calibrate* method **(D)** | ECE | – | 3.44 | 2.99 | 3.17 |
| | Brier | – | 0.415 | 0.410 | 0.406 |

Table 2: CIFAR10: Influence of the Ensemble Size

**Ablation study** : We focus on the CIFAR10 dataset with $N_{train} = 1000$ fixed training examples, and 100 different validation sets of size $N_{val} = 50$: Table 3 reports the means and standard deviations across these experiments. For setups involving training a single model, we report the mean and standard deviations of the metric from a variety of 30 different trained models.

| Metric | (Ours) 30 models Method **[D]** Augment + mixup | 30 models mixup Augment | single model mixup Augment | single model no mixup Augment | single model no mixup no Augment |
|---|---|---|---|---|---|
| test acc | $69.92 \pm .04$ | **70.67** | $66.45 \pm .61$ | $63.73 \pm .51$ | $49.85 \pm .66$ |
| test ECE | $\mathbf{3.3} \pm 1.9$ | 13.9 | $7.03 \pm .7$ | $20.7 \pm .4$ | $23.4 \pm 1.0$ |
| test NLL | $\mathbf{0.910} \pm .012$ | 0.961 | $1.03 \pm .13$ | $1.509 \pm .017$ | $1.770 \pm .045$ |
| test BRIER | $\mathbf{0.414} \pm .002$ | 0.431 | $0.463 \pm .005$ | $0.556 \pm .006$ | $0.718 \pm .009$ |

Table 3: Ablation study performed on CIFAR10 1000 samples. For ensemble temp scaling, we use 950 training samples and 50 validation sets. For setups with variation, we report metric mean and standard deviation.

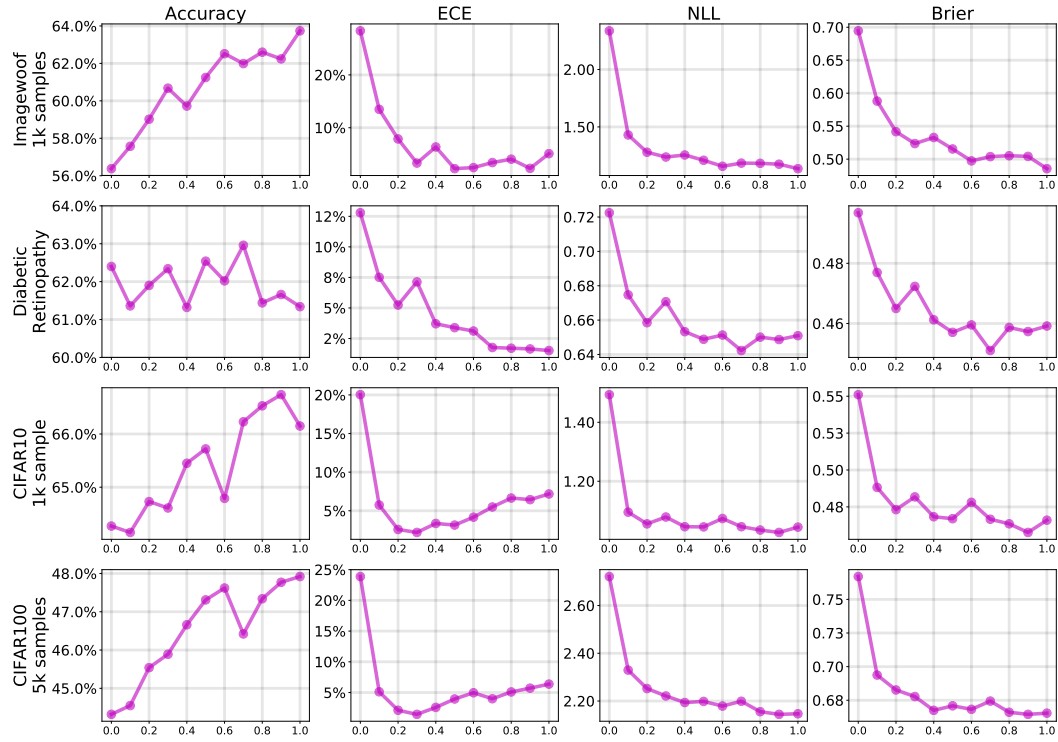

Figure 2: **Mixup $\alpha$ (x-axis) vs. Metric (y-axis)**: The effect of a higher mixup in NLL, ECE, BRIER score is quite evident in the plots. In our setting, most of the metrics improve as a function of $\alpha$. The CIFAR{10,100} datasets show a slight increment in the ECE because the model starts to become under-confident. In contrast, the other three metrics for CIFAR show improvement. From top to bottom, the datasets are Imagewoof 1000 samples, Diabetic Retinopathy with 5000 samples, CIFAR10 with 1000 samples, and CIFAR100 with 5000 samples. The metric in each row is test accuracy, test ECE, test NLL, and test Brier from left to right.

**Effect of mixup $\alpha$** In figure 2 we list generalization and calibration results of high $\alpha$ mixup augmentation. All the setups in which we analyze the performance are limited in the number of training data points. It shows that even if with adequate data, high mixup makes models under-confident; for low data settings, mixup with $\alpha$ near 1.0 boosts model performance quite significantly.

**Detailed numerical results** In table 4 we present the detailed numerical results for all our setups. The table includes result of our proposed *Pool-then-calibrate* method **[D]**, the vanilla pooling method **[A]**, and that of the individual models. The conclusions are consistent across all the setups.

| Method | Test Accuracy | Test ECE | Test NLL | Test Brier |
|---|---|---|---|---|
| CIFAR10 - 1000 samples | | | | |
| Single model | $66.48 \pm .62$ | $7.31 \pm .7$ | $1.037 \pm .013$ | $0.464 \pm .005$ |
| Vanilla pooling **[A]** | 70.71 | 13.9 | 0.961 | 0.431 |
| Pool-then-calibrate **[D]** | 70.71 | $4.9 \pm 2.9$ | $0.916 \pm .015$ | $0.417 \pm .005$ |
| CIFAR100 - 5000 samples | | | | |
| Single model | $46.8 \pm .41$ | $5.4 \pm .37$ | $2.180 \pm 0.014$ | $0.674 \pm 0.003$ |
| Vanilla pooling **[A]** | 55.32 | 17.8 | 1.911 | 0.623 |
| Pool-then-calibrate **[D]** | 55.32 | $2.1 \pm .5$ | $1.787 \pm .002$ | $0.592 \pm .0$ |
| Diabetic Retinopathy (5000 samples) | | | | |
| Single model | $61.26 \pm .62$ | $2.96 \pm .64$ | $0.657 \pm 0.004$ | $0.465 \pm 0.004$ |
| Vanilla pooling **[A]** | 64.38 | 4.9 | 0.641 | 0.450 |
| Pool-then-calibrate **[D]** | 64.38 | $2.9 \pm .8$ | $0.637 \pm .002$ | $0.446 \pm .001$ |
| Imagenette (1000 samples) | | | | |
| Single model | $78.67 \pm .34$ | $14.45 \pm .95$ | $0.796 \pm 0.012$ | $0.332 \pm 0.005$ |
| Vanilla pooling **[A]** | 80.91 | 18.2 | 0.753 | 0.312 |
| Pool-then-calibrate **[D]** | 80.91 | $3.5 \pm 1.0$ | $0.638 \pm .005$ | $0.273 \pm .001$ |
| MNIST (500 samples) | | | | |
| Single model | $89.3 \pm .8$ | $6.4 \pm .9$ | $0.375 \pm .022$ | $0.163 \pm .01$ |
| Vanilla pooling **[A]** | 90.53 | 8.4 | 0.351 | 0.151 |
| Pool-then-calibrate **[D]** | 90.53 | 2.1 | 0.306 | 0.139 |

Table 4: Numerical result of Vanilla pooling **[A]** and Pool-then-Calibrate **[D]** for different setups. In our chosen setups, the pooled predictions are consistently more under-confident than single models. *Pool-then-calibrate* has the best performance across all the metrics.

## 2 Deviation from Calibration

To obtain a more quantitative understanding of why ensembles are under-confident, we consider a binary classification framework. For a pair of random variables $(X, Y)$, with $X \in \mathcal{X}$ and $Y \in \{-1, 1\}$, and a classification rule $p : \mathcal{X} \rightarrow [0, 1]$ that approximates the conditional probability $p_x \approx \mathbb{P}(Y = 1 | X = x)$, we define the *Deviation from Calibration* (DC) score as

$$\text{DC}(p) \equiv \mathbb{E}\left[\left(\mathbf{1}_{\{Y=1\}} - p_X\right)^2 - p_X(1 - p_X)\right]. \tag{1}$$

The term $\mathbb{E}\left[\left(\mathbf{1}_{\{Y=1\}} - p_X\right)^2\right]$ is equivalent to the Brier score of the classification rule $p$ and the quantity $\mathbb{E}\left[p_X(1 - p_X)\right]$ is an entropic term (i.e. large for predictions close to uniform). Note that DC can take both positive and negative values and $\text{DC}(p) = 0$ for a well-calibrated classification rule, i.e. $p_X = \mathbb{E}\left[\mathbf{1}_{\{Y=1\}} | p_X\right]$ almost surely. This can be obtained by observing that

$$\text{DC}(p) = \mathbb{E}\left[\mathbf{1}_{\{Y=1\}} - p_X\right] + 2 \cdot \mathbb{E}\left[p_X^2 - p_X \cdot \mathbf{1}_{\{Y=1\}}\right] \tag{2}$$

$$= \mathbb{E}\left[\mathbb{E}\left[\mathbf{1}_{\{Y=1\}} - p_X \,\middle|\, p_X\right]\right] + 2 \cdot \mathbb{E}\left[\mathbb{E}\left[p_X^2 - p_X \cdot \mathbf{1}_{\{Y=1\}} \,\middle|\, p_X\right]\right]$$

$$= \mathbb{E}\left[\mathbb{E}\left[\mathbf{1}_{\{Y=1\}} \,\middle|\, p_X\right] - p_X\right] + 2 \cdot \mathbb{E}\left[p_X^2 - p_X \cdot \mathbb{E}\left[\mathbf{1}_{\{Y=1\}} \,\middle|\, p_X\right]\right],$$

and then putting $p_X = \mathbb{E}\left[\mathbf{1}_{\{Y=1\}} | p_X\right]$. Furthermore, among a set of classification rules with the same Brier score, the ones with less confident predictions (i.e., larger entropy) have a lesser DC score.

In summary, the DC score is a measure of confidence that vanishes for well-calibrated classification rules, and that is low (resp. high) for under-confident (resp. over-confident) classification rules. Contrarily to the entropy functional, the DC score is extremely tractable.

For a set of $K \geq 2$ classification rules $p^{(1)}, \ldots, p^{(K)}$ and non-negative weights $\omega_1 + \ldots + \omega_K = 1$, the linearly averaged classification rule $\bar{p} := \sum_{i=1}^{K} \omega_i \, p^{(i)}$ satisfies

$$\mathrm{DC}\left(\sum_{i=1}^{K} \omega_i \, p^{(i)}\right) = \sum_{i=1}^{K} \omega_i \, \mathrm{DC}\left(p^{(i)}\right) - \underbrace{\sum_{i,j=1}^{K} \omega_i \omega_j \, \mathbb{E}\left[\left(p^{(i)} - p^{(j)}\right)^2\right]}_{\geq 0}. \tag{3}$$

Equation (3) shows that averaging classifications rules decreases the DC score (i.e. the aggregated estimates are less confident). Furthermore, the more dissimilar the individual classification rules, the larger the decrease. Even if each model is well-calibrated, i.e., $\mathrm{DC}(p^{(i)}) = 0$ for $1 \leq i \leq K$, the averaged model is not well-calibrated as soon as at least two of them are not identical. The derivation in (3) can be obtained by the following steps -

$$\begin{aligned}
\mathrm{DC}(\bar{p}) &= \mathbb{E}\left[\left(\mathbf{1}_{\{Y=1\}} - \bar{p}\right)^2\right] - \mathbb{E}\left[\bar{p} \cdot (1 - \bar{p})\right] \\
&= \sum_{i,j=1}^{k} \omega_i \omega_j \mathbb{E}\left[\left(\mathbf{1}_{\{Y=1\}} - p^{(i)}\right)\left(\mathbf{1}_{\{Y=1\}} - p^{(j)}\right)\right] - \sum_{i,j=1}^{k} \omega_i \omega_j \mathbb{E}\left[p^{(i)}\left(1 - p^{(j)}\right)\right] \\
&= \sum_{i,j=1}^{k} \omega_i \omega_j \, \mathbb{E}\left[\mathbf{1}_{\{Y=1\}} - p^{(i)} - \mathbf{1}_{\{Y=1\}} p^{(i)} - \mathbf{1}_{\{Y=1\}} p^{(j)} + 2 p^{(i)} p^{(j)}\right] \\
&= \sum_{i=1}^{k} \omega_i \mathbb{E}\left[\mathbf{1}_{\{Y=1\}} - p^{(i)} + 2 \cdot p^{(i)}\left(p^{(i)} - \mathbf{1}_{\{Y=1\}}\right)\right] - \sum_{i,j=1}^{k} \omega_i \omega_j \, \mathbb{E}\left[\left(p^{(i)} - p^{(j)}\right)^2\right] \\
&= \sum_{i=1}^{k} \omega_i \mathrm{DC}(p^{(i)}) - \sum_{i,j=1}^{k} \omega_i \omega_j \, \mathbb{E}\left[\left(p^{(i)} - p^{(j)}\right)^2\right]
\end{aligned}$$