# OpenReview forum: "Uncertainty Quantification and Deep Ensembles"
_NeurIPS.cc/2021/Conference — NeurIPS 2021 Poster_

### Official Review · Reviewer_LouX · 2021-07-16

**Rating:** 6
**Confidence:** 4

**Summary:**

This paper empirically studies the calibration of deep ensembles in low-data settings, as well as the interaction between ensembling, post-hoc calibration (i.e. temperature scaling) and mixup data augmentation (as a form of regularization). The main experimental findings include that in the low-data regime, 1) ensembling neural networks (i.e. averaging their predictions) _reduces_ their confidence, 2) mixup regularization generally makes neural networks _underconfident_, 3) as a result, ensembling mixup-regularized models does _not_ improve calibration but instead exacerbates their underconfidence, and 4) temperature scaling can help improve the calibration of deep ensembles, but only if it is applied _after_ averaging the individual model predictions.


**Limitations And Societal Impact:**

Yes, the paper describes the limitations of the work by clearly stating the assumptions under which the presented results hold. It also addresses its potential societal impact in sufficient detail.


**Main Review:**

**Post Rebuttal**

The author response clarifies some of the issues I had raised -- I therefore increase my score from weak rejection (5) to weak acceptance (6). I am still hesitant to recommend acceptance more enthusiastically due to the remaining issue of a low methodological contribution.

---

This paper tackles an important open problem in practical deep learning (which is of significant interest to the NeurIPS community), namely how to obtain models that are well calibrated and therefore robust. It provides a comprehensive and well-executed empirical study with conclusive results and actionable suggestions that will be useful to deep learning practitioners. The manuscript is also very well-written and easy to follow. That being said, the methodological contribution and novelty is rather low, and the scope of the evaluation somewhat limited (by focusing on the low-data regime). Importantly, while the empirical results and resulting practical suggestions are certainly interesting and useful, previously published work has made very similar observations. In particular, it has been shown in prior work that Mixup + ensembling deteriorates calibration [1] (which the authors acknowledge, claiming that it succeeds their work) and that temperature scaling after ensemble averaging improves calibration [2] (which the authors seem to have missed). While this seems to compromise the novelty of this paper, I can certainly understand the authors claims that the work was conducted in concurrence (at least [1]), so I am somewhat torn. All in all, I view this paper as borderline, and am tending to recommend rejection — but I am happy to change my judgement and increase my score if the authors can convincingly address my concerns.


Detailed comments are as follows:

- The methodological contribution is rather low; the paper does not introduce a new method, but rather conducts a study to assess the interaction between different existing approaches, namely deep ensembling, Mixup regularization/augmentation and temperature scaling; the lack of a methodological contribution is not an issue in itself, but raises the bar for the empirical evaluation.
- The empirical evaluation is generally well-executed and fairly comprehensive. It studies the interaction of practically relevant and widely-used techniques (ensembling, mixup augmentation & temperature scaling) in a variety of settings. The main conclusions are convincingly supported by the results and provide clear suggestions for practitioners. Also, the paper is generally very well written and enjoyable to read.
- While the results are certainly interesting and useful, the main issue I see with this work is that previously papers (which have been published already) have made very similar observations. In particular:
     - [1] also studied the interaction between ensembling and Mixup, reaching the same conclusion that they both individually reduce confidence, and that combining them therefore yields compounded underconfidence which deteriorates calibration. [1] then continues to argue that this is caused by the conflation of model and data uncertainty, and proposes CAMixup, which improves upon Mixup by adaptively adjusting the mixing parameter on a class-wise basis. The paper clearly mentions [1] in their related work section (l. 294-297), but claims that it is "subsequent to our work". I personally cannot comment on that, and would suggest the authors to clarify this directly with the area chair.
    - [2] has a different goal to this paper ([2] studies issues with commonly-used calibration metrics and suggests to always use temperature scaling for a fair comparison of model calibration), but their empirical evaluation ultimately ends up with the same recommendation as this work, namely that applying temperature scaling on top of the ensemble predictive is beneficial to calibration. I appreciate that the paper under review has a different focus and also does a more comprehensive evaluation of the combination between temperature scaling and ensembling (in particular, assessing different ways/orders of combining the two), but this nevertheless seems to impact the novelty and potential significance of this work. I would appreciate the author's comments on this. It might be worth pointing out that the paper does not mention [2] anywhere, and that [2] was published at ICLR 2020, so it would likely not be considered as concurrent work.
- One could argue that by focusing on the small-data regime, the scope of the evaluation becomes somewhat narrow. However, I believe that the small-data regime is important and therefore welcome studies focusing on it, as most other papers typically look at the large-data regime. That being said, it is clear that some of the conclusions drawn from this work do not translate beyond the small-data setting, which perhaps could be clarified even more. For example, the authors acknowledge that for large datasets, the differences in performance between the methods assessed are relatively minor (l 291-293). It would be great if the authors could clarify what exactly they mean by that (ideally by showing the larger-scale experiments that were conducted to reach this conclusion).


Questions:
- In l.115-118 you state that „in the low-data regime, […] we have empirically observed that […] Vector and Matrix scaling post-processing calibration methods do not offer any significant advantage over the simple and robust temperature scaling approach.“; I did not find those experiments in the paper or appendix, were these just preliminary results that you did not intend to share? I believe that many people would be interested in seeing this comparison (me included), so it might be worth in including it in the appendix.
- It might be clearer to describe the setup and implementation details (l. 285-293) at the beginning of the experiments, rather than at the end.
- Some of the figure captions just describe the plots, but do not state the conclusions drawn from them. While I understand that this is personal preference, I personally find it easier to parse if the conclusions are summarized in the captions, to avoid having to jump back-and-forth between the text and figures.
- In Fig. 2, for the unscaled ensemble, why is mixup with alpha=0.2 much more confident than the rest (e.g. when looking at entropy)?
- Typo in l.249: „As regards the“ —> „With regards to the“ / „Regarding the“
- The plot labels/legends are not consistent (compare e.g. Fig 2 and Fig 3/6, where the mixup parameters are labeled in different ways).
- In l. 287, you mention that you took the 5th place solution from the Kaggle challenge for the Diabetic Retinopathy dataset; it would be great to have a justification for this.
- Typo in l. 311: „Designing“ —> „designing“
- In the references, paper [2] and [3] are identical and should be merged into a single reference.



[1] Wen et al. 2021, "Combining Ensembles and Data Augmentation can Harm your Calibration" (https://arxiv.org/abs/2010.09875).

[2] Ashukha et al. 2020, "Pitfalls of In-Domain Uncertainty Estimation and Ensembling in Deep Learning" (https://arxiv.org/abs/2002.06470).


**Time Spent Reviewing:**

7

---

> ### Author Response · Authors · 2021-08-10
> **related work + large-data regime + vector/matrix scaling**
>
> We thank the reviewer for their detailed comments (and very clear summary of our contribution!). The positive remarks from the reviewer are encouraging. Below we address some of the concerns raised by the reviewer.
>
> **Comparison with other works:** As mentioned by the reviewer, the work [2] has a different goal altogether and does not focus on the issue of under-confidence of the predictions or when the ensemble actually ends up worsening calibration issues. However, [2] is indeed very relevant and it will be discussed in the final version. As per the reviewer's suggestion, we have clarified the concurrency of the mentioned work to the Area chairs. For work [1], without compromising anonymity, we would like to point out that we first uploaded (an essentially identical version of) our work in July 2020 (arxiv stamp) -- if the Area Chair could discuss these issues (and confirm the timestamps + content), that would certainly be very helpful. We are now in the awkward (but not uncommon) situation when other manuscripts cite our work, convey similar observations, and are accepted in (good) venues. To the best of our knowledge, our manuscript is the very first to have raised several points that the reviewer has very clearly summarized.
>
> **Extension to large-data setting:** In lines 291-293 of the main text, we initially meant that when data is plentiful, the gains offered by most methods are usually significantly less impressive than in the low-data regime, as studied in the text. Nevertheless, we thank the reviewer for the comment and we now believe that we have been a bit fast with this claim. Although classification accuracy is usually not an issue when data is plentiful (and most methods perform about the same), the lack of calibration can indeed be still present when models are trained with aggressive data-augmentation strategies (as is common nowadays): the distributional shift between (data-augmented) training samples and (non-augmented) test samples when models are used in production can lead to significant calibration issues. We now believe our method to still be relevant to the large-data regime (although training large ensembles can be computationally unfeasible -- admittedly, the motivations for concentrating on the low data regime were: (1) our lab has limited computational ressources and a proper study of the calibration properties at imagenet-scale is (very) far from being possible (2) most of our applied work in medical imaging is concerned with situations with a small number of labelled samples). We leave these larger-scale experiments for further work due to computational constraints and thank the reviewer for this remark. As a preliminary step in that direction, we have investigated below the CIFAR100 (full dataset/ ResNet architecture / no-mixup).
>
> | Method                              |    Accuracy    |      ECE      |      NLL      |     Brier     |
> | ----------------------------------- | :------------: | :-----------: | :-----------: | :-----------: |
> | (1) Individual models (unscaled)    | 70.8% &pm; .36 | 9.8% &pm; .31 | 1.17 &pm; .01 | 0.411 &pm; .0 |
> | Ensemble of models in (1)           |     78.4%      |     5.9%      |     0.782     |     0.308     |
> | (2) Individual models (temp scaled) | 70.8% &pm; .36 | 2.1% &pm; .4  | 1.07 &pm; .01 | 0.396 &pm; .0 |
> | Ensemble of models in (2)           |     78.4%      |     13.2%     |     0.859     |     0.331     |
> | Pool-then-calibrate                 |     78.4%      |     3.4%      |     0.770     |     0.303     |
>
> The first row reports the performance of individual models trained without mixup: the individual models are over-confident, but not extremely over-confident (presumably because of the large number of samples). When these models are pooled to make an ensemble in the second row, the pooled model is somewhat calibrated. *This* is the setup that is usually studied  in almost every early articles investigating the properties of deep-ensembles, hence leading to the (somewhat misleading) conclusion that deep-ensembling inherently brings calibration. When we make the individual models calibrated in the 3rd row (here we used temp-scaling but it can also be due to the effect of more aggressive data-augmentation schemes), the individual calibration naturally improves significantly. Nevertheless, when we pool these calibrated models to make an ensemble, the pooled model suffers from extreme under-confidence (4th row). Our proposed method (5th) row performs well (and is straightforward to implement).
>
> **Matrix and Vector scaling:** We have performed (and initially not reported) these experiments. Because of the small size of the validation set (~50 samples in our experiments), Matrix/Vector often overfitted the validation set. For example, for the CIFAR10 dataet (ie. 10 classes), vector/matrix scaling approaches need to tune 10/100 parameters respectively. Below, we tabulate the results we have obtained for vector/matrix scaling when used for the CIFAR10 dataset in the low-data setting (mixup parameter 1.0 / 1000 training samples / 50 validation samples). These results will be added to the Suplementary Material.
>
> | Method              | Accuracy |  ECE  |  NLL  | Brier |
> | ------------------- | :------: | :---: | :---: | :---: |
> | Temperature scaling |  70.71%  | 4.9%  | 1.037 | 0.464 |
> | Vector scaling      |  63.14%  | 12.4% | 1.534 | 0.505 |
> | Matrix scaling      |  63.08%  | 12.5% | 1.536 | 0.505 |
>
> **Implementation details and choice of architecture for Diabetic Retinopathy:** Thank you for this suggestion, details will be added after the "Background" section for better readability. For the "Diabetic Retinopathy" dataset, we choose the mentioned solution as it was used in the serious and well-writen Uncertainty Quantification study [3], and the code was available on Github. Furthermore, the neural architecture was significantly different from the other models used in the manuscript (a concern raised during a previous submission of our work).
>
> **Figure captions and inconsistency of labels**: This will be fixed in the final version of the manuscript, as well as the several typos raised by the reviewer: thank you!
>
>
> **Reference:**
> [1] Wen et al. 2021, "Combining Ensembles and Data Augmentation can Harm your Calibration" (https://arxiv.org/abs/2010.09875).
>
> [2] Ashukha et al. 2020, "Pitfalls of In-Domain Uncertainty Estimation and Ensembling in Deep Learning" (https://arxiv.org/abs/2002.06470).
>
> [3] Leibig et al.  2017, "Leveraging uncertainty information from deep neural networks for disease detection" (doi = {10.1038/s41598-017-17876-z})

---

> > ### Comment · Reviewer_LouX · 2021-08-24
> > **Updated score to recommend acceptance (6)**
> >
> > Thank you very much for your detailed and insightful response -- I particularly appreciate the additional experimental results!
> > I have confirmed with the Area Chair and Wen et al. 2021 is indeed concurrent work.
> > Your response clarifies some of the main reservations I had -- I will therefore update my score to recommend weak acceptance (6).
> > My concern of the low methodological contribution still remains, which is the main reason I am hesitant to more enthusiastically recommend acceptance (i.e. 7).

---

> > > ### Author Response · Authors · 2021-08-26
> > > **"low methodological contribution still remains"**
> > >
> > > Thank you very much for your comments and for updating you rating -- we believe that your requested additional simulations will significantly improve the quality of the manuscript.
> > >
> > > As regards your comment on the "low methodological contribution still remains", in order to emphasise the important practical consequences of the proposed method, would it help if we listed a subset of the large number of papers that do study the use of deep-ensembles for enhanced Uncertainty Quantification and robustness. We would like to inquire whether the "low methodological contribution" refers to the fact that the proposed method is extremely simple to implement (which, to our view, is a big advantage). If the proposed method were significantly more complicated and enjoyed similar performances, would the reviewer assessment remain the same?

---

> > > > ### Comment · Reviewer_LouX · 2021-08-31
> > > > **Thanks for your response!**
> > > >
> > > > Thank you for your response and for further commenting on my criticism of a lack of a strong methodological contribution.
> > > >
> > > > I would like to clarify that I certainly agree that the simplicity of the proposed approach (i.e. applying temperature scaling to calibrate the ensemble average post-hoc) is an advantage rather than a disadvantage. I am also aware of the large corpus of recent works that use deep ensembling methods to improve the uncertainty calibration of neural networks, and hence the importance of any approaches improving ensemble calibration.
> > > >
> > > > The main issue I see is that this exact approach had already been proposed in a previous paper [1] that you seemed to have missed, as I had pointed out in my review and as you had acknowledged in your response. Despite the different focus and approach of the two papers, the resulting method is exactly the same, which in my view takes away from your paper's methodological novelty. While I have no doubt that you came up with the approach independently, [1] had unfortunately already been published at ICLR 2020 and is therefore considered to be prior work in my understanding (I can certainly empathise that this is unfortunate and frustrating -- I am sorry that I had to point this out). Again, this only impacts the methodological novelty/contribution and not any other aspect of your paper.
> > > >
> > > > [1] Ashukha et al. 2020, "Pitfalls of In-Domain Uncertainty Estimation and Ensembling in Deep Learning" (https://arxiv.org/abs/2002.06470).

---

> > > > > ### Author Response · Authors · 2021-09-01
> > > > > **Thank you.**
> > > > >
> > > > > Thank you for explaining further your comments, it is now clearer what was exactly meant.
> > > > >
> > > > > Your suggestions have been very useful to us: thank you for the time you have spent reading our manuscript, and for the overall pleasant, efficient and worthwhile review process.

---

### Official Review · Reviewer_KgKq · 2021-07-17

**Rating:** 6
**Confidence:** 4

**Summary:**

An empirical analysis of the interplay between deep ensembles, data augmentation and calibration with temperature scaling. The paper seeks to clear out several misconceptions about ensembling (eg. ensembling does not necessarily imply better calibration -- this is only true if each network in the ensemble is overconfident) and to provide practical recommendations (eg., order in which the ensemble aggregation and the calibration should be carried out).

**Limitations And Societal Impact:**

Well covered overall

**Main Review:**

**Overall appreciation (originality, quality, clarity, and significance)**
- Clarity: A well motivated and very well written paper for which the main claims would benefit from more thorough / rigorous proofs to be more convincing. The paper is very easy to read throughout -- all key terms are adequately defined, the work is adequately structured and flows naturally
- Quality: The work draws conclusions and provides guidance based on (sometimes limited) empirical evidence only (no theoretical proof; empirical analyses limited to 4 datasets, single model architecture for each). Some aspects of the experimental design might be problematic (see below).
- Originality: The interplay between ensembling, calibration and data augmentation has been understudied to date -- this paper provides some interesting elements of response
- Significance: limited to a couple insightful takeaways (the one about calibration vs aggregation ordering needs some clarification)

**Clarifying questions / suggestions**
- Introduction → Please clarify what you mean by “low data regime” -- this may mean different things to different people
- Section 4 → In the descriptions of all experiments, you seem to only reference a training set (on which models are trained) and a validation set (on which the temperature scaling is tuned) but no test set. Are all results reported in Fig2-5 reported on the same validation set on which you perform the temperature scaling tuning? If so, the claim that “pool-then-calibrate” is better than “calibrate-then-pool” approach seems trivial. Conclusions drawn in section 4 (Fig5 in particular) would only be meaningful if results are reported on a separate test set
- Section 4, line 200 → Did you look into other types of aggregation (e.g., geometric or harmonic mean)?
- Related work (line 294-297) → This “related work” section feels a bit contrived as such. Would suggest expanding a bit more

**Minor points**
- Line 41 -- adapting (typo)
- Line 49 -- augmentations (typo)
- Line 180 -- drop “a” after “as well as” (typo)
- Line 292-293 -- “When datasets are large, the differences in calibration performances between  the several methods are relatively minor” -- please cite
- Line 311 -- drop capital letter for “Designing”

**Time Spent Reviewing:**

7

---

> ### Author Response · Authors · 2021-08-10
> **important clarification on the use of "test sets" + alternate pooling**
>
> We sincerely thank the reviewer for their constructive suggestions and feedback. We also appreciate the reviewer’s effort in pointing out the several typos (now corrected). Below, we address some of the concerns raised by the reviewer.
>
> **"limited to a couple insightful takeaways"**: Our two main contributions are:
> 1. the realization and empirical demonstration that the ensembling process does not lead in general to better calibration properties (as is very often claimed in the literature), but instead lower the confidence of the individual network (which is only useful when each network is over-confident, which is far from being the case with modern data-augmentation techniques) -- although simple (** in hindsight **), we believe that it is a very important property that, to the best of our knowledge, has not been studied before our contribution.
> 2. we propose a simple methodology to mitigate the above-described issues. In a range of experiments, we have demonstrated that the methodology is robust, simple to deploy and can halve the ECE score. As a matter of fact, we are now using this simple methodology in most of our applied project in medical imaging (labelled data is scarce and uncertainty quantification is an important concern to medical practitionners).
>
> **"the one about calibration vs aggregation ordering needs some clarification"**:
> We advocate the approach "(D) Aggregate first the estimates of each individual model before calibrating the pooled estimate" and empirically demonstrate that calibrating before averaging the predictions leads to significantly less robust results. As a matter of fact, we are now using this simple methodology in most of our applied project in medical imaging (labelled data is scarce and uncertainty quantification is an important concern to medical practitionners).
>
> **Low data regime:** thank you for the suggestion to clarify this term. We refer to "low data" regime the situation when overfitting is an especially important concern and when the addition of new labelled samples would significantly improve the classification accuracy (ie. the model is still far from having reached the regime when additional data provides negligible gains in robustness/accuracy). We choose this setting for various reasons.
>
> 1. It is indeed very common in practice (industrial/medical) -- "big-data" is a luxury in most industries.
> 2. Data-augmentation strategies and ensembling, as studied in the text, are especially important in this regime in order to mitigate overfitting. In the "big-data" regime, it is fair to say that most of these techniques only lead to modest gains, contrarily to the low-data regime where these methods are essential.
> 3. Ensembling is especially simple to deploy since it is computationally inexpensive to train models in the low-data regime.
>
> **Choice of validation and test dataset:** For all our experiments the test set is fixed and unseen during the whole training and validation (temperature scaling) procedures. We keep the test set as the original provided test set for all the datasets. We choose our “low-data” training set by randomly choosing a subset of the original training set. Afterwards, we choose a validation dataset by randomly choosing a subset from the remaining leftover training dataset. We perform our experiments on different such choices of validation sets in order to include error bars. For all the experiments requiring a validation set (e.g. temperature scaling) we perform all the post-processing optimization on the validation set (keeping the validation same for all competing methods), and then finally evaluate all the methods on the unseen test dataset (different from the validation). For example, for CIFAR10, the original training dataset contains 50k examples and the test dataset contains 10k examples. For our setting, we first choose 1k examples from the training dataset as our training data, and then choose 50 validation examples from the remaining 49k training data. The test dataset (10k) is used as is (and never used during the training phase)
>
> **Alternate pooling strategies:** We have also tried several alternate pooling strategies such as percentile pooling (min, max, median), geometric pooling. We did not include the results of these pooling strategies because of space constraints in Figure 5 of the main text. We have observed that the resulting performances associated to Geometric Pooling were essentially similar to linear pooling, and hence mainly concentrated on linear pooling (because more intuitive and more common). On the other hands, the harmonic-mean pooling strategy was not stable and led to consistently worse results. For completeness, we will now add these results in the Supplementary Material. Thank you for this important suggestion.
>
> **Related work:** We purposedly shortened the related work due to space constraints. This section will be expanded in the final version (with 1 additional page allowed). Thank you.

---

> > ### Comment · Reviewer_KgKq · 2021-08-26
> > **Thank you for the response**
> >
> > Dear authors,
> >
> > Thank you very much for the detailed response. To summarize my thoughts about the paper:
> >
> > 1. It exposes the fact that model ensembling may not necessarily translate to better calibration, in particular when ensembling low-confidence models -- a common misconception which would be worthwhile for the larger community to be more broadly aware of.
> > 2. It provides practical guidance on how to address this issue (i.e. ensemble models then do temperature scaling) and shows benefits of the suggested approach in the low data regime.
> > 3. While empirical evidence is provided for both, theoretical and methodological contributions of the paper are overall limited.
> >
> > As for that last point, I want to clarify that this has nothing to do with the simplicity of the suggested approach -- which I would argue is a strength rather than a weakness of the approach. But the methodological contributions are limited to practical guidance on the ordering of ensembling and temperature scaling, and empirical evidence is limited to experiments in the low data regime.
> > With that said, given the responses you provided during the rebuttal cleared out doubts I had about the experiment design, and because I do believe the ideas developed in the paper are well exposed and promote good practice, I have increased my score to weak accept.

---

> > > ### Author Response · Authors · 2021-08-27
> > > **on the large-data regime**
> > >
> > > Dear Reviewer,
> > >
> > > Thank you very much for careful reading of our manuscript and for your insightful comments and clarification requests that have helped us improve the exposition and description of the proposed method.
> > >
> > > We wanted to briefly comment on the low/large data regime. Although classification accuracy is usually not an issue when data is plentiful (and most methods perform about the same), the lack of calibration can indeed be still present when models are trained with aggressive data-augmentation strategies (as is common nowadays): the distributional shift between (data-augmented) training samples and (non-augmented) test samples when models are used in production can lead to significant calibration issues. We believe our method to still be relevant to the large-data regime (although training large ensembles can be computationally unfeasible -- admittedly, the motivations for concentrating on the low data regime were: (1) our lab has limited computational resources and a proper study of the calibration properties at imagenet-scale is (very) far from being possible (2) most of our applied work in medical imaging is concerned with situations with a small number of labelled samples). We leave these larger-scale experiments for further work due to computational constraints. As a preliminary step in towards studying the large-dataset setting, we have investigated below the CIFAR100 (full dataset/ ResNet architecture / no-mixup).
> > >
> > > | Method                              |    Accuracy    |      ECE      |      NLL      |     Brier     |
> > > | ----------------------------------- | :------------: | :-----------: | :-----------: | :-----------: |
> > > | (1) Individual models (unscaled)    | 70.8% &pm; .36 | 9.8% &pm; .31 | 1.17 &pm; .01 | 0.411 &pm; .0 |
> > > | Ensemble of models in (1)           |     78.4%      |     5.9%      |     0.782     |     0.308     |
> > > | (2) Individual models (temp scaled) | 70.8% &pm; .36 | 2.1% &pm; .4  | 1.07 &pm; .01 | 0.396 &pm; .0 |
> > > | Ensemble of models in (2)           |     78.4%      |     13.2%     |     0.859     |     0.331     |
> > > | Pool-then-calibrate                 |     78.4%      |     3.4%      |     0.770     |     0.303     |
> > >
> > > The first row reports the performance of individual models trained without mixup: the individual models are over-confident, but not extremely over-confident (presumably because of the large number of samples). When these models are pooled to make an ensemble in the second row, the pooled model is somewhat calibrated. This is the setup that is usually studied in almost every early articles investigating the properties of deep-ensembles, hence leading to the (somewhat misleading) conclusion that deep-ensembling inherently brings calibration. When we make the individual models calibrated in the 3rd row (here we used temp-scaling but it can also be due to the effect of more aggressive data-augmentation schemes), the individual calibration naturally improves significantly. Nevertheless, when we pool these calibrated models to make an ensemble, the pooled model suffers from extreme under-confidence (4th row). Our proposed method (5th) row performs well in this large-is data regime (and is straightforward to implement).

---

> > > > ### Comment · Reviewer_KgKq · 2021-08-31
> > > > **On the large-data regime**
> > > >
> > > > Dear authors,
> > > >
> > > > Thank you for the additional clarification. I fully agree with you that calibration issues may also be a concern in the large-data regime (e.g., due to data-augmentation). My comment was on the nature of the evidence presented in the paper, which focused on the low-data regime.

---

### Official Review · Reviewer_SiXE · 2021-07-18

**Rating:** 6
**Confidence:** 4

**Summary:**

This paper focuses on the calibration performance of deep ensembles of over-parameterized models. In particular, this work studies the interaction between data augmentation, ensembling and post-processing calibration such as temperature scaling. Each of the technique improves calibration when applied alone. However, the authors demonstrated that both mixup and temperature scaling might hurt calibration when naively combined with ensembling. The empirical evaluation identifies the underconfidence as the core issue in calibration degradation. It also shows that the proposed modification to temperature scaling can greatly improve calibration.

**Limitations And Societal Impact:**

The authors discussed the limitation.

**Main Review:**

Both mixup and temperature scaling are prevailing methods to improve deep neural networks calibration. However, their interactions with deep ensembles received only little attention. This paper is a step towards understanding these interactions. It identifies that the compounding underconfidence is the core issue of worse calibration. Temperature scaling is generally used without extra caution but the authors showed that it need to be tweaked when combined with ensembels. In particular, the order of pooling and scaling affects the calibration of deep ensembles and the authors found that the Pool-Then-Calibrate approach gives the best result.

I also have several concerns, one of the major concern is that the paper focuses on the empirical interaction between mixup, temperature scaling and deep ensembles. However, both mixup and temperature scaling are studied before (although not with deep ensembles). The fix proposed by this paper is tweaking the order of pooling and temperature scaling rather than any fundamentally different idea. Although some new insights are discovered, the algorithmic contribution is still limited.

The empirical evaluation is limited in the vision domain. The paper would be more convincing if it can show its conclusion can be extended to transformer models in the NLP domain. Moreover, one of the important benefits of ensembles is it can improve uncertainty estimation on out-of-distribution dataset. The paper can also be convincing if the authors can show consistent conclusion on CIFAR-10C and CIFAR-100C dataset.

The authors didn't state which neural network architectures are studied (it is in the appendix saying 30 ensembles of ResNet-18). It would be interesting if the paper can study the case for a small number of ensembles such as 4.

At the current moment, the cons outweigh the pros so I consider the paper is marginally below the boarderline.

**Time Spent Reviewing:**

3

---

> ### Author Response · Authors · 2021-08-10
> **Expanded experiments + Clarifications on contributions + cf. Supplementary Material**
>
> We thank the reviewer for his valuable comments and suggestions. We have ran additional simulations to adress some of his concerns also have clarified a few points:
>
> **Contribution:**
> "The fix proposed by this paper is tweaking the order of pooling and temperature scaling rather than any fundamentally different idea. Although some new insights are discovered, the algorithmic contribution is still limited." Thank you. Our two main contributions are:
> 1. the realization and empirical demonstration that the ensembling process does not lead in general to better calibration properties (as is very often claimed in the litterature), but instead lower the confidence of the individual network (which is only useful when each network is over-confident, which is far from being the case with modern data-augmentation techniques) -- although simple (**in hindsight**), we believe that it is a very important property that, to the best of our knowledge, has not been studied before our contribution.
> 2. we propose a simple methodology to mitigate the above-described issues. The simplicity of the proposed approach should not, we believe, be mistaken for a "limited" contribution. In a range of experiments, we have demonstrated that the methodology is robust, simple to deploy and can halve the ECE score. As a matter of fact, we are now using this simple methodology in most of our applied project in medical imaging (labelled data is scarce and uncertainty quantification is an important concern to medical practitionners).
>
>
> **Out-of-distribution performance:** We show the performance (difference between the median of in-class and out-of-class) of our method compared to vanilla ensembling in the Supplementary Material. **In Table 1 of the Supplementary Material** we show the performance of the two methods when the ensembles are trained on CIFAR10 and tested on a subset of CIFAR100 classes which are visually different from CIFAR10. For convenience, the table is reproduced below. It shows the difference between the medians of the in-class and out-of-class prediction entropy (higher is better).
>
> | Single model 30 variations | Deep Ensemble method[A] | Pool-then-calibrate method [D] |
> | :------------------------: | :---------------------: | :----------------------------: |
> |      0.342 &pm; 0.015      |          0.359          |             0.521              |
>
> Pool-then-Calibrate performs very significantly better than vanilla ensemble in separating the predictions for in-class and out-of-class observations (45% more separation in terms of distance between medians). As suggested by the reviewer, we also run inference on the **CIFAR10-C** dataset after training our ensemble model on the setting: 1000 samples of CIFAR10 dataset with mixup 1.0. CIFAR10-C has five corruption severity, with 5 being the highest level of image corruption. Due to constraints in formatting, we only report one corruption "*Gaussian Noise*" and only highest severity 5. We can add more results related to this in the final version.
>
> | Method | Accuracy | ECE  | NLL  | Brier |
> | ------ | :------: | :--: | :--: | :---: |
> | *Single models* (30 variations) | 59.38% &pm; 0.05 | 6.57% &pm; 0.006 | 1.237 &pm; 0.012 | 0.549 &pm; 0.005 |
> | *Deep Ensemble* (method **[A]**) | 64.63% | 15.13% | 1.145 | 0.511 |
> | *Pool-then-Calibrate* (method **[D]**) | 64.63% | 1.65% | 1.059 | 0.480 |
>
> **It can be seen that the same conclusion holds for CIFAR10-C dataset as well**. The single models are indeed less under-confident than the ensemble itself. Pool-then-calibrate brings has the best result.
>
> **Architectures:** In the main text at **line 285** in the paragraph “*Setup and implementation details*” we discuss in detail the architectures used for each of the dataset we have used. For CIFAR 10/100 we use ResNet18, ResNet34 for Imagenette/Imagewoof, and for the Diabetic Retinopathy, we use the architecture (not containing any residual connection) from the 5th place solution of the associated Kaggle challenge. We chose the prevailing architectures for each dataset and chose three different kinds of architectures, one of them not even convolutional (LeNet on MNIST).
>
> **Varying number of models in Ensemble:** In **Figure 1** of the *Supplementary*, we empirically show the effect of the ensemble size on our conclusion. We also tabulate the detailed numerical results for varying ensemble sizes (including 4 models) in **Table 2** of the *Supplementary*. We show that the conclusion is consistent across different numbers of models in the pool. We include a snapshot of the mentioned table here. The table contains the ECE when the model is trained on the setting CIFAR10, 1000 samples with mixup. All the details are presented in the Supplementary Material. We report the mean ECE for different possible choices of the validation set for the second row (i.e., pool-then-calibrate).
>
> | Method \ Number of models               |   4    |   8    |   15   |
> | :-------------------------------------- | :----: | :----: | :----: |
> | *Vanilla Ensemble* (method **[A]**)     | 12.37% | 13.44% | 13.87% |
> | *Pool-then-calibrate*  (method **[D]**) | 3.44%  | 2.99%  | 3.17%  |

---

> > ### Comment · Reviewer_SiXE · 2021-08-18
> > **The additional experiments  partially addressed my concerns**
> >
> > Thanks for submitting the author response!
> >
> > The additional out-of-distribution experiements did address some of my concerns. And the author responses addressed the concerns about architectures and varying number of ensmeble members. My concern about algorithmic contribution still remains. Therefore I raise my score to 6.

---

> > > ### Author Response · Authors · 2021-08-26
> > > **"algorithmic contribution still remains"**
> > >
> > > Thank you very much for your comments and for updating you rating.
> > >
> > > We do indeed agree with your assessment that "The fix proposed by this paper is tweaking the order of pooling and temperature scaling". In order to emphasize the importance of this (algorithmically simple) remark, would it help if we listed a subset of the large number of papers that do study the use of deep-ensembles for enhanced Uncertainty Quantification and robustness. We find it slightly unfair to judge our contribution based on the fact that it is simple to implement (although it leads to consistent and significant gains in robustness on a method that is widely used and that has existed for years). If the proposed method were significantly more complicated and enjoyed similar performances, would the reviewer assessment remain the same?

---

### Decision · Program_Chairs · 2021-09-27

**Decision:**

Accept (Poster)

**Comment:**

This paper studies the quality of the uncertainty of ensembles of deep networks through the lens of calibration.  The authors demonstrate that contrary to common belief, ensembling can lead to even worse calibration.  They identify that an altered version of temperature scaling, a common strategy, can help to significantly improve the calibration of these ensembles.  During the discussion period a question came up regarding novelty compared to an existing published paper that was very related.  Given the timeline of that paper compared to this one, the reviewers were instructed to treat that paper as concurrent work.

There was significant discussion for this paper during the author response period, multiple reviewers raised their scores and a consensus of weak accept was reached.  Quoting from the discussion, the reviewers all "agreed that 1) the paper studies an important problem (i.e. the interplay between ensembling, data augmentation and post-hoc calibration) that is of high relevance and interest to to the NeurIPS community, 2) the empirical evaluation fairly convincingly supports the key hypotheses (with some minor caveats in terms of breadth of experimental settings), and 3) the resulting practical suggestions (e.g. to apply post-hoc calibration after ensembling) are interesting and useful. Furthermore, there were no concerns raised w.r.t. the writing, which was clear and of high quality."  However, the reviewers agreed that there is not a strong strong algorithmic, methodological, or theoretical contribution and thus no reviewers were willing to champion the paper with a rating higher than a marginal accept.

The fact that all reviewers were unwilling to recommend an "accept" rating or champion this paper seems to indicate that while they believe it meets the bar for acceptance, they don't think it will be tremendously impactful in its current form.  This appears to be partially owing to the fact that the paper has been on arxiv for a considerable time and in the meantime other papers have reported similar results.  Therefore, the recommendation is to accept the paper as a poster.  The paper could be more impactful if there was some additional theoretical justification for the observed phenomenon or some additional analysis, e.g. from the loss landscape perspective.